# Managing Animal Welfare in Food Governance in Norway and Sweden: Challenges in Implementation and Coordination

**DOI:** 10.3390/ani11071899

**Published:** 2021-06-25

**Authors:** Frida Lundmark Hedman, Frode Veggeland, Ivar Vågsholm, Charlotte Berg

**Affiliations:** 1Department of Animal Environment and Health, Swedish University of Agricultural Sciences, P.O. Box 234, SE-532 23 Skara, Sweden; lotta.berg@slu.se; 2Department of Health Management and Health Economics, Norwegian Institute of Bioeconomy Research (NIBIO), University of Oslo, P.O. Box 1089, Blindern, 0317 Oslo, Norway; frode.veggeland@medisin.uio.no; 3Department of Biomedical Sciences and Veterinary Public Health, Swedish University of Agricultural Sciences, P.O. Box 7028, SE-750 07 Uppsala, Sweden; ivar.vagsholm@slu.se

**Keywords:** animal transport, animal welfare, horizontal coordination, management, official control, public administration, slaughter, vertical coordination

## Abstract

**Simple Summary:**

Animal welfare is an important issue in society, and having a strong animal welfare legislation is per se important. However, in addition to a strong legislation, it is necessary to create a system that can enforce the legislation and to have a public administration in place in order to achieve a coordinated implementation. Both Norway and Sweden have received some criticism for their coordination of animal welfare control efforts. However, they have reacted to this criticism in different ways. Norway has centralised the coordination, making the Norwegian Food Safety Authority (NFSA) solely responsible for animal welfare control. Sweden, on the other hand, has instead focused on developing better guidelines to be used by the 21 regional County Administration Boards in order to improve uniformity. In this study, we have compared the Norwegian and Swedish ways of coordinating animal welfare control and identified challenges and relevant organisational preconditions for achieving uniform and consistent compliance. The results show that Sweden’s organisation may need more coordination between multiple organisational units, while Norway has better preconditions for achieving uniformity in animal welfare administration. However, in Norway, the safeguards for the rule of law might be an issue, due to NFSA acting as de facto “inspector”, “prosecutor” and “judge”.

**Abstract:**

A key issue in food governance and public administration is achieving coordinated implementation of policies. This study addressed this issue by systematically comparing the governance of animal welfare in Norway and Sweden, using published papers, reports, and legal and other public information, combined with survey and interview data generated in a larger research project (ANIWEL). Governing animal welfare includes a number of issues that are relevant across different sectors and policy areas, such as ethical aspects, choice of legal tools, compliance mechanisms and achieving uniform control. Based on the challenges identified in coordinating animal welfare in Norway and Sweden, relevant organisational preconditions for achieving uniform and consistent compliance were assessed. The results showed that Sweden’s organisation may need more horizontal coordination, since its animal welfare management is divided between multiple organisational units (Swedish Board of Agriculture, National Food Agency and 21 regional County Administration Boards). Coordination in Norway is managed solely by the governmental agency Norwegian Food Safety Authority (NFSA), which has the full responsibility for inspection and control of food safety, animal health, plant health, as well as animal welfare. Thus, Norway has better preconditions than Sweden for achieving uniformity in animal welfare administration. However, in Norway, the safeguards for the rule of law might be an issue, due to NFSA acting as de facto “inspector”, “prosecutor” and “judge”.

## 1. Introduction

Animal welfare is attracting considerable attention from researchers, political decision-makers, NGOs and consumers [1,2,3,4,5,6,7,8]. There is a growing consensus that animal welfare is ethically important in its own right, irrespectively of how the animal is used by humans. This paper focuses on one particular element of animal welfare management, namely its role in food governance. Thus, when referring to animal welfare, we are in fact primarily referring to food animal welfare. A key issue in studies of public administration and food governance is how to achieve coordinated implementation and application of policies that cut across different levels of government and across different policy sectors and public agencies [9,10,11,12]. This issue was addressed in this study by systematically comparing the public administration systems in place to manage animal welfare in Norway and Sweden. Over the last few decades, both Norway and Sweden have made substantial changes in their systems in order to improve coordination. However, they have chosen completely different strategies, and hence, a systematic comparison may also be useful to policymakers in other countries, before making similar decisions and choosing an approach for improving coordination.

Animal welfare involves different parts of the agri-food governance system, including government agencies, levels of government, farms, slaughterhouses, and transport and lairage systems. A crucial part of animal welfare management is the protection of companion animals, which, in many countries, is subject to a different governance framework than the protection of farm animals. There are also separate international rules and guidelines for companion animals such as the European Convention for the Protection of Pet Animals (signed by both Norway and Sweden). In Norway and Sweden, companion animals are, however, subject to the same basic regulatory animal welfare framework as farm animals, although both countries have adopted some rules and policing measures that are specifically aimed at companion animals. This paper presents some statistics that include companion animals (because parts of the official statistics do not separate between farm and companion animals), but will not consider the companion animal welfare legislation in any detail because it is outside the food chain, i.e., the production of food of animal origin regulated by EU Reg. No. 2017/625. 

Public management of animal welfare poses multiple challenges related to coordination of issues, which are relevant across different sectors and policy areas, such as uniform inspection and controls, enforcement measures, compliance and ethical aspects [4,13,14,15,16,17,18,19]. These challenges are reflected in the European Union (EU) legislation on control of animal welfare, e.g., Regulation (EU) No. 2017/625 of the European Parliament and Council, on official controls and other official activities performed to ensure the application of food and feed law, rules on animal health and welfare, plant health and plant protection products. The Regulation requires Member States to “ensure efficient and effective coordination between all authorities involved, and the consistency and effectiveness of official controls”. Thus, one of the core challenges of animal welfare management is to establish effective coordination. Moreover, the present comparative study of public management of animal welfare gave insights on enabling and constraining factors for achieving consistent and coordinated public policies.

This study analysed in particular: the degree to which public management systems for animal welfare in Norway and Sweden differ; how well-coordinated these two systems are; and how key characteristics of the systems may affect the potential for coordinated, consistent and uniform policy implementation. The ways in which the Norwegian and Swedish administration systems deal with the challenge of coordinating animal welfare were compared. Managing animal welfare related to transportation to slaughterhouses will be used as an example to illuminate some striking differences. The basic assumption for the study is that well-coordinated systems are better apt to ensure implementation and realisation of animal welfare objectives of the national legislation.

## 2. Analytical Framework

The analytical framework (the concepts of vertical and horizontal coordination) was derived from the literature on governance, public management and administration, and organisation theory [9,10,12,20,21,22,23]. Coordination is deemed as being at the core of government activity, not least because of the need to achieve coherent action and policy outputs [11,12,24], i.e., the need to “contribute jointly to—or at least not undermine—specific objectives” [25]. In this paper, emphasis is put on the assessment of the capacity of the Norwegian and Swedish management systems to coordinate action and policies between different public agencies and levels of governance (state, regional, local) in order to realize stated animal welfare objectives. Coordination can relate to an end-state, i.e., to what degree actions/policies are characterized by “minimal redundancy, incoherence and lacunae” [11], and to a process, i.e., the design of structures, lines of communication, etc., which are directed towards achieving a coordinated end-result. Thus, a coordinated animal welfare administration (process) is assumed to enhance coordinated animal welfare policies/measures (end result). Moreover, the underlying assumption is that well-coordinated animal welfare policies will enable decision-makers to more effectively meet animal welfare policy objectives and gain confidence from stakeholders and the general public. Two particular dimensions were deemed relevant for categorisation and analysis: 1) The horizontal or fragmented–coordinated dimension, i.e., the extent to which authority and responsibilities are dispersed across different sectors and public agencies; and 2) the vertical centralised–decentralised dimension, i.e., the extent to which authority and responsibilities are coordinated and mandated by higher levels of governance. Public management systems may be more or less coordinated horizontally (fragmentation/coordination dimension) and vertically (centralisation/decentralisation dimension). For example, strong coordination mechanisms at higher levels of government, e.g., at the national or EU level, are considered to be centralising vertical coordination elements, whereas strong coordination mechanisms between different public agencies, or even allocation of responsibilities to or within a single agency, are considered horizontal coordination elements. A variety of different mechanisms can contribute to well-coordinated policies. However, the underlying assumption in this study was that well-coordinated, integrated systems characterised by a combination of strong vertical and horizontal coordination capacities are more likely to enable uniform enforcement and compliance with public policies than fragmented, decentralised systems. Thus, coordinated animal welfare administrations are assumed to be in a better position to coordinate action in order to realize animal welfare objectives stated in law.

## 3. Material and Methods

This study was part of a larger research project on animal welfare management and governance in Norway (ANIWEL) that involved analyses of a wide variety of qualitative and quantitative data, collected through interviews, field work, desk-top studies and surveys. The present study used those data to identify key coordination issues, which are important for ensuring uniform and consistent implementation of animal welfare legislation. Some of the data are presented and discussed in Gezelius, 2019 [26]. This study was, however, primarily a desk-top study based on qualitative analysis of the information available, such as secondary literature, reports, legal and other public documents [27]. This was supplemented with data from previous studies on animal welfare management in Norway [28,29] and Sweden [13,17,19,30]. This study falls broadly into the field of comparative public administration [31], where a key question is how to organize public administration in order to ensure effective implementation of public policies.

## 4. Governance of Animal Welfare in Norway and Sweden

### 4.1. Norway’s Animal Welfare Administration

#### 4.1.1. Background

In the early 1990s, the Norwegian animal welfare administration was dispersed across both different agencies and levels of government [32], i.e., it was vertically decentralised and horizontally fragmented. The public meat inspection (at slaughterhouses) was carried out by local food control authorities. In 1996, this responsibility was transferred to a central agency, the Norwegian Food Control Authority (NFCA). (The NFCA, established in 1988, was given responsibility for food inspections, but not animal welfare and control of plants and aquatic animals. In 2004, the NFCA became part of NFSA.) This transfer was necessary to ensure, e.g., formalised and centralised responsibility for meat control on Norway joining the European Economic Area (EEA) Agreement, to comply with EU’s internal market legislation. (The EEA Agreement covers most EU legislation on the four freedoms (free movement of goods, people, capital and services) including food control and animal welfare. However, Norway is not a part of the EU’s customs union, Common Agricultural Policy (CAP) or Common Fisheries Policy (CFP).)

Under the EEA Agreement, Norway is required to implement most of the EU’s food safety and veterinary regulations, including on animal welfare. In the early 1990s, animal welfare inspections at the farm level were performed by county veterinarians, who were part of regional state administrations run by the county governors [32]. In 1996, these responsibilities were transferred to the newly established state-level agency, the Norwegian Animal Health Authority. However, the local food control authorities continued to perform animal welfare and meat inspections and controls at slaughterhouses, based on authority delegated by the NFCA. Then, in 2004, all responsibilities for plant health, animal health, animal welfare, and food safety inspection and control (including the NFCA and the local food control authorities) were amalgamated into the newly established governmental agency—Norwegian Food Safety Authority (NFSA). Thus, in 2004 all responsibilities for animal welfare inspection and control (and food safety, animal health and plant health inspection and control) at the farm and slaughterhouse level were centralised to one competent authority, the NFSA. One aim of this reform was to improve horizontal and vertical coordination, i.e., to ensure uniform and consistent control across different regions and targets of inspection.

#### 4.1.2. Allocation of Authority and Responsibilities across Levels of Governance (Vertical Coordination)

Two levels of government are involved in the management of animal welfare in Norway: the EU/EEA level and state level. In addition, several administrative levels are involved within the state level of government (two ministries, one state agency, and one central and one regional administrative level within the state agency). The Norwegian government is responsible for initiating and implementing animal welfare legislation adopted by the Norwegian parliament. However, this legislative process is guided and informed by EU legislation adopted at the EEA level. Norway applies ministerial rule, i.e., individual ministers can interfere with details within their respective area of competence and can be held responsible by Parliament, which can even force individual ministers to resign.

Between 2004 and 2015, the NFSA was organised into three administrative levels: state, regional and local. Since February 2015, only two administrative levels remain—the central level (headquarters in Oslo) and regional level (five administrative regions). Each regional level is organised into approximately 30 offices at different locations within the regions.

#### 4.1.3. Allocation of Authority and Responsibilities across Agencies and Sectors (Horizontal Coordination)

Two sector ministries share the formal responsibility for animal welfare in Norway: the Ministry of Agriculture and Food (primarily terrestrial animal welfare), and the Ministry of Trade, Industries and Fisheries (primarily aquatic animal welfare) [33]. However, the Ministry of Agriculture and Food is the main competent ministry for animal welfare and has the lead administrative responsibility for the NFSA (e.g., budgetary control). These ministries oversee and manage the NFSA with regard to activities within their respective jurisdiction and areas of responsibility. They coordinate the NFSA’s follow-up of government animal welfare policies and activities aimed at providing input to the EU, and thereby EEA-relevant legislative work on animal welfare. However, this set-up creates a challenge for horizontal coordination at the government level.

Within the NFSA, horizontal coordination challenges arise with regard to ensuring a uniform enforcement of animal welfare legislation across regions and inspection offices within these regions. One example arises when the NFSA detects non-compliance with animal welfare legislation in transport and/or farming during inspection on arrival at the slaughterhouse [34]. There is then a need to achieve effective coordination and communication between NFSA inspectors located at the slaughterhouses and the NFSA department/s responsible for inspecting the farms of origin. Thus, horizontal coordination is essential in order to achieve effective enforcement. To enhance horizontal coordination, the NFSA has developed a number of guidelines to achieve uniformity of practices, both between and within regions. It also relies on bottom-up coordination activities by organizing regular coordination meetings for regional directors and seminars where representatives from different regions meet to discuss and calibrate animal welfare activities, application of animal welfare regulations, how to perform inspections, and the use of enforcement tools. Designated advisors on animal welfare in each region are responsible for providing up-to-date knowledge on animal welfare and for coordinating and advising on animal welfare issues within the region. However, because the advisors have no ultimate decision-making authority, the effectiveness of this coordination activity is uncertain. Horizontal coordination challenges, thus, arise in the relationship between two ministries, between the five regions of the NFSA, and between the inspection units within these regions (Figure 1).

#### 4.1.4. Animal Welfare Inspections in Norway

The NFSA carries out different types of inspections including risk-based standard inspections, random (ad hoc) standard inspections and acute inspections based on public complaints [34]. Animal welfare inspections in Norway are normally carried out by veterinarians, many of whom are also responsible for performing other types of inspections, such as food safety inspections. It is therefore difficult to specify the exact number of inspectors dealing with animal welfare. When assessing compliance with Norwegian animal welfare legislation, the Norwegian government must also consider the need to comply with the animal welfare legislation within the EEA Agreement and the potential for action by the EFTA Surveillance Authority (ESA) and eventually the EFTA Court, in cases of non-compliance. Thus, Norway has a well-specified system for enforcement of animal welfare legislation, but the organisation of the system poses challenges with regard to achieving uniform and consistent practices across the five administrative regions and the offices within these regions.

### 4.2. Sweden’s Animal Welfare Administration

#### 4.2.1. Background

During recent decades, Sweden has implemented major changes in its animal welfare administration. In 2004, responsibilities for animal welfare and the role of central competent authority (CCA) were relocated from the Swedish Board of Agriculture (SBA) to the newly established Swedish Animal Welfare Agency (SAWA) [13]. The reason was that the SBA was considered to have too many tasks and sometimes conflicting roles in the management of food and agriculture issues [35]. For example, in addition to animal welfare, the SBA was, and still is, responsible for improving the competitiveness of national agriculture, and for increasing the production of Swedish food [36,37]. However, in July 2007, the role of CCA was returned to the SBA, mainly because of the new government’s desire to make one agency responsible for all questions concerning agriculture [38]. Another important reform was made in 2009, when responsibilities at the operational level, i.e., for conducting animal welfare inspections and official controls, were moved from the municipal level to regional level, i.e., to the County Administrative Boards (CABs). The main reason was to increase the standardisation and uniformity within animal welfare control in Sweden, i.e., to improve horizontal coordination [39].

#### 4.2.2. Allocation of Authority and Responsibilities across Levels of Governance (Vertical Coordination)

As of 2021, the public responsibility for animal welfare in Sweden is dispersed vertically on three levels of government: EU, state and regional level [40]. Sweden is required to comply with EU legislation, and as a Member State, is audited and inspected by the DG Health and Food Audits and Analysis (formerly known as the Food and Veterinary Office, FVO), to ensure that EU legislation is properly implemented and enforced. The Swedish government (i.e., the Ministry of Enterprise and Innovation) is responsible for providing the SBA with instructions, assignments and the budget concerning the priorities and tasks of the SBA. However, Sweden does not apply ministerial rule, which means that its national authorities are more independent compared to the Norwegian authorities.

The SBA writes the national regulations concerning housing, management, transport and slaughter of different kinds of animals and activities. However, they also provide guidance for the CABs to achieve uniform implementation of legislation among different parts of Sweden [40,41]. To achieve this, the SBA runs an Animal Welfare Council where they, together with the CABs and the National Food Agency (NFA), shall develop the official animal welfare control to make it more legally secure, equitable and effective (Chapter 8, Section 11, the Swedish Animal Welfare Ordinance). The SBA provides the CABs with checklists and guidelines to support animal welfare inspectors. The SBA has also developed a risk classification model for CABs to use when planning and calculating the control frequency for different animal premises [42].

#### 4.2.3. Allocation of Authority and Responsibilities across Agencies and Sectors (Horizontal Coordination)

Since Sweden is divided into 21 counties (regions), there are 21 CABs carrying out official animal welfare controls [40]. This clearly highlights the challenge of horizontal coordination in Sweden, i.e., the challenge of achieving uniform and consistent practices across different regions responsible for animal welfare management. In order to improve uniformity, the CABs have animal welfare coordinators who meet on a regular basis, and the section heads responsible for animal welfare inspectors have formed a network. An additional challenge regarding horizontal coordination is that the public responsibility for managing animal welfare at the farm level (the CABs) is separated from the public responsibility for managing animal welfare at slaughterhouses. For slaughterhouses, the official veterinarians of the NFA are in charge of everyday animal welfare inspections and controls, but the CABs are still responsible for animal welfare audits and follow-ups at slaughterhouses, especially in cases of non-compliance. Thus, there is a strong need for horizontal coordination of animal welfare management in multiple organisation units: NFA, SBA and 21 CABs (Figure 2).

#### 4.2.4. Animal Welfare Inspections in Sweden

The CABs carry out different types of inspections: risk-based standard inspections, random (ad hoc) standard inspections, acute inspections based on public complaints or cross-compliance inspections [43]. The latter cover requirements originating from EU legislation, and cross-compliance failure can lead to a reduction in EU transfers to the farmer. The animal welfare inspectors usually have a university degree in biology (e.g., Ethology and Animal Welfare), environmental health or animal science. In 2018, there were 218 animal welfare inspectors in Sweden [44]. Each CAB also employs at least one official veterinarian as part of the animal welfare group. Even if Sweden has a well-specified system for enforcement of animal welfare rules, its animal welfare management system poses some challenges with regard to uniform and consistent practices, mainly because multiple agencies and inspectors (SBA, NFA, 21 CABs) are involved in actual application and enforcement of the legislation.

## 5. Methods Used to Enforce Compliance and Punish Non-Compliance in Norway and Sweden

### Description of Enforcement Methods and Tools

The methods and tools used to uncover and sanction non-compliances in Norway and Sweden are quite similar, but there are a few differences (Table 1). Both Norway and Sweden use the escalating scale principle according to the principle of proportion when non-compliance is discovered [26,45]. The logic of the escalating scale principle in regulatory governance is that “every escalation of non-compliance by the firm can be matched with a corresponding escalation in punitiveness by the state” [46]. Both Norway and Sweden refer to this principle in their manuals for implementing sanctions in cases of non-compliance with animal welfare rules. Depending on the severity of non-compliance and the animal keeper’s willingness to change and comply with the legislation, different tools can be used. A formal notice (inspection report) is the mildest tool and is not a decision or a sanction per se, and therefore, cannot be appealed. An injunction can be issued if the non-compliance is more severe or repeated, or if the animal keeper refuses to make voluntary corrections. An injunction is a binding decision that can be appealed. It can also be linked to a conditional fine, i.e., a financial cost, so that a lower cost is involved for the animal keeper to make the prescribed corrections than to pay the fine. The conditional fine has to be imposed by a court order, not by the CAB or NFSA, but the state. In Norway, the NFSA can also impose an administrative fee on individuals or firms for infringements that have already taken place. This type of administrative fine is based on administrative law, and not on criminal law. Sweden, being a member of the EU, uses cross-compliance. Norway has a similar system of cross-compliance, but based on domestic agricultural policy, i.e., the reduction in subsidies is decided by the Norwegian agricultural authorities.

In both Norway and Sweden, the authorities have the legal power to seize animals if the welfare situation is poor, or if previous non-compliances resulting in injunctions have not been rectified. The authorities can also force the animal owner to sell or give away the animals within a certain time frame (i.e., obligation to terminate a holding of animals). Both the NFSA and the CAB can also impose a ban on keeping or handling animals. This decision is guided by type of non-compliance and the risk of future maltreatment or neglect of other animals. It is important to acknowledge that seizing possessions and issuing a ban is not a penalty, but an administrative tool based on civil law in order to stop and prevent current and future severe animal welfare problems and suffering.

In both countries, the authorities can report to the judicial system (police/prosecutor) any non-compliance with animal welfare legislation they may detect. The judicial system must then investigate whether a crime has been committed, whether the animal owner should be prosecuted and set a penalty. The punishment in Sweden for violating animal welfare legislation is a fine or maximum two years’ imprisonment. An animal owner in Sweden can also be convicted of “animal cruelty” if the animal has suffered. Such severe offences are regulated under the penal code, and not the Animal Welfare Act, but the potential punishment is the same, although there is currently a suggestion for adding a stricter penalty for very severe cases [47]. In Norway, the punishment is a fine or maximum one year’s imprisonment, but for serious violations, the penalty is imprisonment for a maximum of three years.

In both Norway and Sweden, animal welfare cases are tried in the general courts. If an administrative decision (e.g., an injunction or administrative fee) is appealed, an administrative court handles the case in Sweden. Norway does not have an administrative court system, so appeals are heard by the central office of the NFSA, i.e., the appeal on a decision made by the regional NFSA is handled by the central level of the NFSA. Decisions made by the central level are final, but may be referred to the general courts or to the Ombudsman. The threshold for taking cases to the general courts is quite high, because of legal costs and because of the need to provide justification for the courts to accept the case.

In 2018, the NFSA detected non-compliance with Norwegian animal welfare legislation on 3063 animal premises, which represented 39% of the 7857 premises inspected (Table 2) [48]. According to the Swedish control statistics from 2018, non-compliances were found in 6402 (57%) of 11,918 inspections [44]. Of 4011 animal premises that received a planned standard inspection from CAB, non-compliances were detected on 1864 (46%).

The most common action taken when non-compliance was detected in Sweden and Norway in 2018 was a formal notice. The use of injunctions was more common in Norway than in Sweden, but seizure of animals was more common in Sweden (Table 2). In all, 82% of Swedish decisions to seize animals involved dogs and cats, while 5% involved farm animals. The figure was similar in Norway, where 89% of seized animals were pets [48].

## 6. Managing Animal Welfare during Transport

### 6.1. Common Ground

Both Sweden and Norway comply with EC Regulation 1/2005 on protection of animals during transport, Sweden as an EU member state and Norway as party to the EEA agreement. According to Regulation 1/2005 Annex 1, “No animal shall be transported unless it is fit for the intended journey, and all animals shall be transported in conditions guaranteed not to cause them injury or unnecessary suffering.” Transport is, for example, not permitted if animals are unable to move independently, have a severe open wound or uterine prolapse or are in late pregnancy. However, sick or injured animals may be transported if they are slightly injured or ill and transport would not cause additional suffering. Both the farmer and the transporter are responsible for meeting the requirements in the Annex, but when an unfit animal arrives at the slaughterhouse, it is not always clear whom to blame and which sanctions to use. There are challenges for transporters in ensuring that animals do not become injured during transport and in spotting single unfit animals when many animals are loaded into a transport vehicle simultaneously. Transporters argue that they are not always able to detect all unfit animals and have to trust the producer not to send unfit animals for transport. (Interview with Norwegian transporters of animals 2016 (three), 2017 (one) and 2020 (one).) Below, we use this case of the responsibilities of the farmer and transporter regarding animal welfare during transport to compare the functioning of the inspection systems of Norway and Sweden.

### 6.2. Norwegian Approach

In addition to EC Regulation 1/2005, Norway has developed domestic legislation concerning animal welfare during transport. According to this, “no person shall transport animals or cause animals to be transported in a way likely to cause injury or undue suffering to them” (Regulation (NO) No. 925 of 30 June 2014, Article 3). Thus, both transporters and animal keepers are responsible and culpable. The problem is whom to hold responsible in individual cases.

In 2012, the EFTA Surveillance Authority (ESA), on a mission to Norway related to animal welfare during transport, pointed out several problems with the Norwegian system, such as lack of specific penalties related to non-compliances with the requirements in Regulation (EC) No. 1/2005 and lack of instructions/guidance on the possible types of penalties. The challenge for NFSA to collate and analyse information about, and actions against, both transporters and animal keepers not complying was noted during the EFTA Surveillance Mission [50]:

*There is not a systematic national approach ensuring official controls at all stages of transport of animals. Official controls concerning the welfare of animals during transport are almost exclusively carried out at arrival at slaughterhouses facilities, missing controls at place of departure, on roads and on roll-on-roll-off ferries*.

Although one agency (NFSA) was made responsible for inspections at all stages of transport of animals, coordination challenges remained. In 2014, Norway adopted new rules on sanctions (c.f. Regulation (NO) No. 925 of 30 June 2014 on administrative fees), and in 2015, the NFSA reported to the ESA that it had updated its instructions to make sure that “non-compliances related to animal welfare, detected during checks on arrival at the slaughterhouse in connection with ante-mortem control, are communicated within or between local departments to allow follow-up in the farm of origin or place of departure” [34]. For this purpose, the NFSA established an electronic database (MATS) for communication with persons/departments in the regions where farms of origin are located [34]. Thus, the NFSA put in place new horizontal coordination mechanisms to ensure that both transporters and animal keepers are held responsible in cases of non-compliance. These measures contributed to the conclusion in the ESA report from 2018 that “coordination is ensured within the NFSA in relation to official controls on animal welfare during transport in line with Article 4(3) of Regulation (EC) No. 882/2004” [34]. The NFSA also developed guidelines for transport of animals to improve uniform assessments [51]. The guidelines are quite detailed and provide examples of when animals are suitable for transport, how animals with special needs can be transported and when animals must be separated during transport [51].

However, the strengthening of horizontal coordination mechanisms did not deal with the problem of fairness, e.g., whether NFSA’s use of administrative sanctions and penalties against transporters is proportionate in relation to the goal of compliance with animal welfare rules. In fact, Article 139 in Regulation (EU) No. 2017/625 states that a sanction shall be “*effective, proportionate and dissuasive*”. The problem is to define and agree on what is actually a “proportionate” sanction. The introduction of administrative fees has lowered the threshold for state-inflicted sanctions because the NFSA does not have to go through the courts. It has moreover strengthened the perceptions of “unfairness” among transporters because appeals are treated by the same body that inflicts sanctions, i.e., the NFSA [26], something which is perceived as undermining the rule of law.

The NFSA also applies a strict responsibility for the transporters, meaning that it is always the transporter who is responsible for assuring that only animals fit for transport arrive at slaughterhouses. There is, thus, an ongoing debate between Norwegian transporters, who criticise the NFSA for disproportionate and unfair penalties, and the NFSA, which argues that the use of administrative fees has helped in achieving better compliance with the transport-related rules on animal welfare.

### 6.3. Swedish Approach

Sweden has also complemented EC Regulation 1/2005 with domestic regulations and general advice (SJVFS 2019:7) concerning transport of animals. The administrative coordination of animal welfare at slaughterhouses in Sweden creates some challenges. The horizontal coordination of regulations and control guidelines needs to be secured at the state level between the SBA and the NFA. Horizontal coordination is also needed at the regional level between the NFA (OVs) and the CABs. Finally, there must be vertical coordination between the SBA, NFA and CABs.

In 2010, Sweden received an FVO audit concerning animal welfare [52]. The FVO reported that the responsibility of CAB and NFA at slaughterhouses needed to be clarified, that the checklists NFA used for animals arriving at slaughterhouses lacked assessment guidelines, and that CABs were carrying out too few official controls of animal transport. Sweden’s reaction was to issue comprehensive guidelines on assessment of animal welfare at slaughterhouses [53], as a complement to existing guidelines on transportation of live animals [54]. The guidelines nowadays also include examples of when the farmer and/or the transporter is responsible for the animals during transport [53], i.e., not only guidance on when an animal is unfit for transport.

Unlike in Norway, appeals are always handled by administrative courts in Sweden. The Swedish courts assess the plausibility of determining whether an infringement has been committed and who is responsible for each situation. If it is evident that the animals were affected during transport, e.g., injured on interior fittings in the vehicle, asphyxiated due to lack of ventilation, etc., it appears that only the transporter is held responsible by the courts [55,56,57]. The same reasoning applies if animals are hurt by the transporter during loading, e.g., if they use an electric prod in an illegal way [58]. If the animal’s unfit condition was evident to both the farmer and the animal transporter before loading, both are held responsible and convicted [59,60,61,62,63]. However, the level of penalties may differ between the farmer and the transporter if the court decides that one of the involved individuals is more to blame. For example, the farmer may be ordered to pay a higher fine than the transporter [63] or be convicted for animal cruelty, or the transporter may be convicted for contravening the Animal Welfare Act [59,62]. In these cases, the farmer has known the condition of the animal for longer and should have acted earlier, while the transporter may only have been able to inspect the animal briefly.

In some cases, only the farmer is held responsible, mainly when the animal has clearly been suffering long before transport, e.g., when the OV at the slaughterhouse discovers chronically sick or very thin animals [64], pigs with a very large scrotal hernia [65,66] or cattle with deformed horns penetrating the skull tissue [67]. In other cases, prosecuted transporters have been acquitted due to poorly designed loading facilities or conditions on-farm at the time of loading, including too dark or narrow areas and overstocking, as these conditions hamper the ability of the transporter to inspect every animal in a reasonable way before loading [68,69,70].

Several requirements need to be fulfilled for a person to be convicted in court. According to the Swedish Prosecution Authority, 53% of preliminary investigations in this field are dropped, in most cases concerning transportation of animals too late in pregnancy or too sick or injured to be fit for transport [71]. The main reason for dropping such investigations appears to be difficulty in proving whether a farmer/transporter knew, or should have known, the condition of an animal or when an injury occurred; hence, it appears difficult to prove that the act was carried out with “intent” or “neglect/indifference”.

## 7. Discussion

### 7.1. Organisational Structures

Both Norway and Sweden comply with EU/EEA legislation and have adopted laws and regulations that go beyond the minimum EU levels. Both countries also face similar challenges in coordinating activities for development and implementation of EU/EEA animal welfare legislation. Moreover, partly to fulfil the requirements of the EU legislation, both countries need an animal welfare administration system that can enable effective, uniform and consistent implementation and application of animal welfare legislation. However, the two countries have chosen different paths to achieve this.

The countries reacted differently to external criticism (by FVO and ESA) concerning their shortcomings in coordination of animal welfare control. Norway developed and formalized strong coordinated structures both vertically and horizontally, with, e.g., allocation of all responsibilities for animal welfare management to one single state-level agency, the NFSA. The horizontal coordination challenges that remain are primarily within the NFSA (between five administrative regions, and between approximately 30 departments/inspection offices within these regions). There is also a certain need for coordination within government, i.e., between the two Norwegian ministries responsible for terrestrial and aquatic animal welfare legislation, respectively. The Norwegian animal welfare administration is, thus, characterised by internal horizontal coordination challenges.

Sweden responded with extensive guidelines concerning the responsibilities of different authorities and retained a more complicated set of administrative structures, where authority and responsibilities are dispersed across one ministry, two state agencies and 21 regional CABs. The coordination and guidance of the inspectors are diffusely allocated to different bodies. The Swedish animal welfare administration is thus characterised by *external* horizontal coordination challenges, where coordination mechanisms have to cross formal organisational borders. The differences between the Norwegian and Swedish systems for animal welfare management demonstrate that, depending on the issue at hand, different types of coordination instruments may be used.

### 7.2. Enforcement Tools and Level of Compliance

Sweden and Norway have quite similar enforcement tools, with the obvious difference related to the use of administrative fees in Norway. However, introduction of administrative fees in Sweden has been suggested [47]. Norwegian farmers and transporters consider administrative fees to be unfair [26] (This also came up in interviews with farmers and transporters conducted in 2016, 2017 and 2020), as the appeal process is, in their case, managed by the same agency (NFSA) that imposes the fees. Such complaints are substantiated by the fact that appeal decisions, in most cases, end up in favour of the NFSA decisions. There is a conflict between the NFSA’s roles as “prosecutor” and “judge”, a problem better taken care of in Sweden through the separation of tasks between administrative courts and inspection agencies.

Based on the comparison of enforcement tools and number of non-compliances (Table 2), Sweden would appear to have a poorer animal welfare level than Norway, as interpreted from the higher proportion of non-compliances, more commonly occurring seizures of animals, higher proportion of bans on keeping animals and more notifications to the police. However, it is highly possible that these figures reflect differences in the legislation and in enforcement strategies, rather than in animal welfare. The results of animal welfare controls are dependent on a number of factors, such as the content of the legislation, how the control objects are selected and the way of measuring non-compliances (e.g., on individual or group level, the use of thresholds and samples). Furthermore, the way of handling non-compliances, access to guidelines and assessment training sessions, and how the control statistics are reported and presented are also relevant [72].

### 7.3. Different Strategies When Implementing the EU Transport Legislation

The case of transporting animals to slaughterhouses was used to exemplify the differences between the administration structures in Norway and Sweden. Detection of non-compliances on arrival of animals at Norwegian slaughterhouses involves only one agency, the NFSA. Thus, the NFSA is freer to choose among all available tools and to coordinate actions between inspectors at the slaughterhouse and those responsible for controls of primary production at farm level. Detection of non-compliances on arrival of animals at Swedish slaughterhouses formally involves two different agencies, NFA and SBA, by involvement of the OVs employed by NFA to check animals at the slaughterhouse, and the regional CABs responsible for controls of primary production, based on the SBA regulations. Thus, in Sweden, decisions on who has the authority to act and on which tools to use for enforcement need to be coordinated between different agencies, so the Swedish system appears to have more organisational barriers to effective coordination than the Norwegian system.

The dispersal of responsibilities and authority across different agencies in Sweden also risks leading to different “organisation cultures” guiding behaviour, i.e., development of separate beliefs, routines and practices among inspectors. Thus, the system is at higher risk of cultural barriers to coordination and uniform and consistent practices. In 2010, the FVO audited Sweden and found some lack of clarity regarding the responsibilities for animal welfare controls, in particular at small slaughterhouses [52]. In response, Sweden developed more detailed guidelines concerning controls of animal transport and slaughterhouses, under which inspectors must report non-compliances and demand corrections. However, it is not up to the inspectors to determine the legal responsibility, which is left to the courts to decide. Based on the verdicts described above, it is clear that the courts make many trade-offs when handling the question of responsibility.

Even if there are clarifying guidelines, there are still large differences and discrepancies in how official animal welfare controls, in general, are carried out in different parts of Sweden [44,73,74,75]. Wilhelmsson and co-workers [76] reported that pig transport drivers in Sweden are worried about being reported for driving unfit animals as the inspectors at the slaughterhouses make inconsistent assessments. Moreover, uniformity and good practice may depend on how the animal welfare controls are actually organised. Studies on Norwegian animal welfare controls indicate an obvious challenge in horizontal coordination within the NFSA when seeking a satisfactory level of inter-observer agreement [26]. Similar results have been reported for Denmark [77,78]. The use of ambiguous wordings in the legislation, e.g., “*unfit*”, “*normal”* or “*suitable*” [75,79,80,81,82]; unclear guidelines on what is acceptable [28,83,84]; and too few training sessions for inspectors [85] impede uniformity, especially when the regulators are striving for a more goal-oriented legislation [79], which is the case in both Norway and Sweden. Therefore, we argue that achieving a uniform and well-functioning animal welfare administration system is a balancing act that requires a well-coordinated organisation and useful and explicit guidelines, including training sessions, and flexible structures to allow for adoption of best practices in local contexts.

The case of transporting animals to slaughterhouses exemplified the differences between Norway and Sweden in handling non-compliances. Norway implemented the ESA requirements on improving coordination and implementation of animal welfare legislation by introducing new legislation on administrative fees in 2014. Hence, NFSA inspectors have to make an assessment concerning non-compliance and on who should pay the administrative fees, i.e., the authority acts as both “inspector” and “judge” [26]. This may be why Norway seems to impose stricter liability on transporters than Sweden, i.e., it is more important and convenient if NFSA inspectors know, without needing to make any trade-offs, whom to hold responsible for non-compliances. This may be a consequence of the Norwegian effort and willingness to centralise and standardise coordination and assessment of animal welfare. Despite the efforts by the Norwegian authorities to coordinate the inspection system and ensure equal treatment, animal transporters perceive the system of administrative fees to be unfair [26]. Thus, the combination of organisational uniformity within the NFSA and infringement procedures, which excludes assessments and second opinions from outside the NFSA, seems to have some adverse effects on the perceived legitimacy of the control system.

## 8. Conclusions

This study examined how the challenge of coordinating animal welfare is addressed within the Norwegian and Swedish administrative systems, and the relevant organisational preconditions for achieving uniform and consistent compliance with official animal welfare goals. Both countries use several mechanisms to coordinate management of animal welfare rules. Norway has implemented radical reforms to achieve this, primarily by giving a single agency (the NFSA) all responsibilities for animal welfare management. Sweden has relatively strong vertical coordination capacity through EU membership, a single-ministry authority and responsibilities being located at the state level. However, the administrative system is more fragmented horizontally than the Norwegian system, as authority is allocated to two different agencies (NFA, SBA) and 21 regional administrative units (CABs). This way of organising animal welfare management is a risk for organisational and cultural barriers, with negative effects on coordination that might undermine uniform and consistent compliance with animal welfare legislation. We believe that these descriptions and our analyses of the two different approaches and their consequences are also of interest to other countries and regions and possibly also outside the animal welfare control arena, as the choice of administrative pathways is a highly relevant issue in many areas of modern society in order to achieve effective implementation of public policies.

Other challenges may arise when trying to improve animal welfare administration uniformity by making one authority responsible for every step. The case of transport of animals illustrated the differences between the Norwegian and Swedish systems. In Norway, the NFSA is responsible for inspection of transport from the farm to the transport vehicle and to the slaughterhouse. If non-compliance is detected when animals arrive at the slaughterhouse, the transporter is always held responsible, and appeals are handled within the NFSA. In Sweden, two different agencies are involved, and an administrative court handles appeals and allocates the legal responsibility between farmer and transporter, making trade-offs and assessments in every case.

This analysis revealed a number of constraining and enabling organisational factors for achieving effective coordination, implementation and legal security. However, better knowledge is needed on operation of management systems in specific situations, in order to identify and specify the effects of coordination capacity on actual outcomes, i.e., the degree of uniform and consistent compliance with stated policy objectives. It is unlikely that one solution will fit all countries, due to different legal systems, traditions and cultures not only in the area of animal welfare but in general. Such differences may affect the potential for effective compliance, together with organisational structures, available legal instruments, the quality of guidelines and the training of animal welfare inspectors. Hence, the outcome of an EU legislation does not only depend on how it is implemented in national legislation, but how the public administration systems are organised and coordinated. We argue that the quality of animal welfare management is also affected by the degree to which it is perceived “fair” and “just”. In this paper, this point was highlighted by referring to perceptions of “unfairness” among Norwegian animal transporters, caused by the fact that NFSA can impose sanctions without going through the courts and that NFSA itself, not a neutral external appellate body, handles appeals. In this regard, having recourse to legal review by the courts to determine culpability, penalties and administrative sanctions appears to improve the perception of fairness. Moreover, uniformity needs to be combined with a certain degree of flexibility of the inspection system and its tools for enforcement, to adapt its practices to the variety of contexts in which it operates, and thus, its ability to ensure legitimacy and create trust.

## Figures and Tables

**Figure 1 animals-11-01899-f001:**
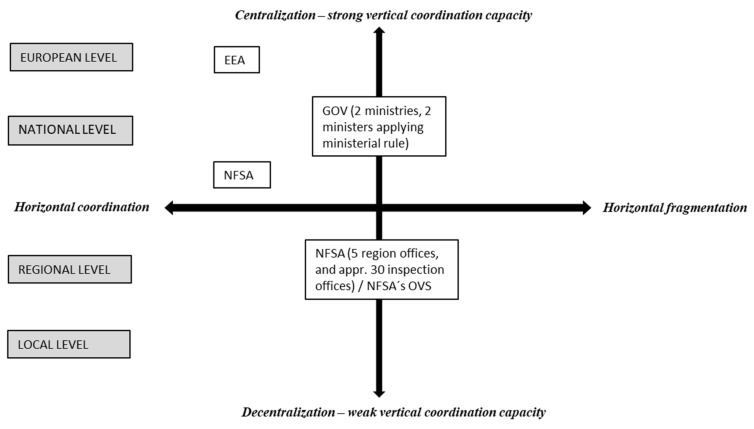
Coordination between public management systems for animal welfare in Norway. EEA = European Economic Area, NFSA = Norwegian Food Safety Authority, OVS = official veterinarian at slaughterhouse.

**Figure 2 animals-11-01899-f002:**
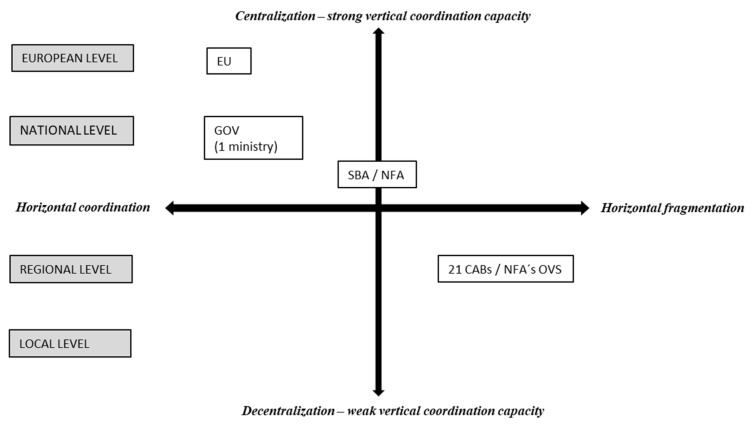
Coordination of public management systems concerning animal welfare in Sweden. SBA = Swedish Board of Agriculture, NFA = National Food Agency, CAB = County Administrative Board, OVS = official veterinarian at slaughterhouse.

**Table 1 animals-11-01899-t001:** Summary of methods used to ensure compliance with the animal welfare legislation and sanctions/punishment for non-compliance in Norway and Sweden. The response escalates from the top down, i.e., the more severe the non-compliance, the harsher the method. Y = yes, N = no.

Method and Tools	Norway	Sweden	Differences between Countries?
Informal methods Guidance—give advice ^a^	Y	Y	N
Formal methods and tools Formal notice Injunction Conditional fine Administrative fee Cross-compliance (reduced subsidy) Animal ban/activity ban Seize animals/wind up holding of animals	Y Y Y Y Y Y Y	Y Y Y N Y Y Y	N N N Y Y ^b^ N N
Criminal tools and sanctions Notification to police/prosecutor Judicial proceedings Legal punishment	Y Y Y	Y Y Y	N Y ^c^ Y ^d^

^a^ The advice given must relate to the legal requirements, and not the practical solutions needed to correct non-compliance. ^b^ Reduced EU subsidy in Sweden, reduced national subsidy in Norway. ^c^ Sweden has an administrative court handling appeals of authority decisions. ^d^ Slightly different sentences.

**Table 2 animals-11-01899-t002:** Number of times or premises on which different tools and actions were used during official animal welfare controls in Sweden and Norway in 2018 (including all animals, e.g., farm animals, horses, pets, etc.). Note that one or several tools/actions were used and applied several times on some animal premises. The two countries sometimes included different types of inspections and types of animal activities in the statistics, so the values are not always completely comparable. NA = not applicable, NK = not known.

	Norway	Sweden
Number of inspected animal premises	7857	NK
Number of inspections	10,797	11,918
Proportion of inspected premises ^a^	11%	6%
Action/tool used		
Injunction	2809 ^b^	1163 ^b^
Conditional fine	168 ^b^	NK
Administrative fee	61 ^c^	NA
Cross-compliance (reduced subsidies) ^d^	73	268
Animal ban/activity ban	159 ^c^	198 ^b^
Seize animals	178 ^b^	895 ^b^
Notification to police/prosecutor	46 ^c^	309 ^b^

^a^ Sweden only reports proportion of farm premises inspected, due to a national goal to inspect at least 10% of farms every year. Norway includes other premises in this number, e.g., horses and pets. ^b^ Number of decisions. ^c^ Number of animal premises. ^d^ For Norway, this number is based on the number of applications for subsidies made to the agricultural authorities where the subsidy was reduced because of breach of animal welfare legislation [49]. NFSA is responsible for informing the agricultural authorities about such breaches. For Sweden, this number is number of farmers that received a reduced EU subsidy due to non-compliance towards EU legislation.

## Data Availability

Not applicable.

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
