# Peer review of "Managing Animal Welfare in Food Governance in Norway and Sweden: Challenges in Implementation and Coordination"

_animals, 2021, doi:10.3390/ani11071899_

Round 1

Reviewer 1 Report

Overall Comments

I enjoyed this paper and believe it is a good fit for animals.

I would  like the paper's title changed as it looks a food governance, that is animal welfare only for the animals we eat, not the animals we keep as pets, for fibre or for work. 

More specifically - The paper talks about animal welfare as a general thing but then looks at food governance. Does the issue that good animal welfare needs to be consistent for all animals irrespective of their human use merit a paragraph or two? We produce animals for food, fibre, fun (companion animals etc), and to work for us. All should be considered in the same way to ensure internal and external consistencies when we consider governance /control / guardianship over animals. 

The paper sees coordination as the key purpose of governance. Ensuring consistent goals and changes flow through networks effectively is also an essential goal of effective governance. You do consider this quite well at lines 109 - 113 but I would like a little more here. Governance is more than just ensuring coordination. Its about meeting policy objectives and stakeholder confidence etc.

Specific Comments 

Line 60 - I am not familiar with the term 'cross cutting' is this a general term?

Line 149 - on farm and slaughterhouse level - perhaps should be at farm and ...

Line 164 - have you previously confirmed what the NFSA is? You define it at line 215 in your Figure 1 but maybe you need to before here?

Line 295 - should define the escalating line principle. Have you considered enough of the basic regulatory theory that looks at compliance models etc?

Line 453 - (and on) I would like this paragraph broken up, its raises a number of interesting things that could sit as separate paragraphs.

Line 629 - I admire your conclusion that a system's effective is enhanced or hindered by perceptions of its fairness and how it creates justice. I cant quite see that you have made this point out strongly enough in the paper to state it as strongly. Are you saying that having access to a court changes people's perceptions?

My Concluding Comment

A well written and considered paper. I would like my comments addressed above but otherwise I recommend this paper.

Author Response

Reviewer: I would like the paper's title changed as it looks a food governance, that is animal welfare only for the animals we eat, not the animals we keep as pets, for fibre or for work.
Our response: The title has been changed to ”Managing Animal Welfare in Food Governance in Norway and Sweden: Challenges in Implementation and Coordination”

Reviewer: More specifically - The paper talks about animal welfare as a general thing but then looks at food governance. Does the issue that good animal welfare needs to be consistent for all animals irrespective of their human use merit a paragraph or two? We produce animals for food, fibre, fun (companion animals etc), and to work for us. All should be considered in the same way to ensure internal and external consistencies when we consider governance /control / guardianship over animals.
Our response: Two sentences on this has been included on page 2 (at the top of the page, line 51-55).

Reviewer: The paper sees coordination as the key purpose of governance. Ensuring consistent goals and changes flow through networks effectively is also an essential goal of effective governance. You do consider this quite well at lines 109 - 113 but I would like a little more here. Governance is more than just ensuring coordination. Its about meeting policy objectives and stakeholder confidence etc.
Our response: New sentence has been included on top of page 3 (line 117-119)

Reviewer: Line 60 - I am not familiar with the term 'cross cutting' is this a general term?
Our response: The term ”cross-cutting issues” is used for issues that are relevant across different sectors and policy areas. In order to avoid misinterpretations of the term, the term has now been replaced by the explanation mentioned above both in the Abstract and on page 2, second paragraph.

Reviewer: Line 149 - on farm and slaughterhouse level - perhaps should be at farm and ...
Our response: This has been changed to ”at farm and…” (line 174)

Reviewer: Line 164 - have you previously confirmed what the NFSA is? You define it at line 215 in your Figure 1 but maybe you need to before here?
Our response: A more detailed description of NFSAis made on page 4, first paragraph, line 171 (where NFSA is firstly mentioned). The Abstract has been changed accordingly.

Reviewer: Line 295 - should define the escalating line principle. Have you considered enough of the basic regulatory theory that looks at compliance models etc?
Our response: Two sentences have been included on page 8 (under 5.1, line 323-327) explaining the escalating scale principle and placing it in the context of regulatory theory by referring to the seminal work of Ayres and Braithwaite on ”Responsive regulation”. This reference has been included in the list of references.

Reviewer: Line 453 - (and on) I would like this paragraph broken up, its raises a number of interesting things that could sit as separate paragraphs.
Our response: More paragraphs have been included to brake up the text.

Reviewer: Line 629 - I admire your conclusion that a system's effective is enhanced or hindered by perceptions of its fairness and how it creates justice. I cant quite see that you have made this point out strongly enough in the paper to state it as strongly. Are you saying that having access to a court changes people's perceptions?
Our response: The issues of fairness is raised, in particular, in the paragraph on page 12 – just before 6.3. A new sentence has been included here to emphasize this point even more. Another sentence is also included in the conclusion to clarify this point.

Reviewer 2 Report

This is a very well written manuscript on a topic that is understudied.  The paper is so tight that I have no concerns or criticism of the content.  My comments are for what is not there.

From page 1-9 the authors are describing “animal welfare” as conceived by EU Reg. No. 2017/625 which deals with animals that will become food.  So for the majority of the paper where the “animal welfare" appears it can be read as “food animal welfare”. At  page 9 . Line 350-367 the subject of “pets” and dogs and cats are included in the narrative. After line 369 we are back to food animals.

The division between pet animal welfare policing and farm animal welfare policing in Anglo-American societies is very substantial (1, 2) a legacy of the delegation of police powers to social actions groups (3).

I would request a paragraph about line 70 that described the quazi-criminal legislation protecting companion animals in the two countries, who provides that policing and how this paper will not consider this protection of animals legislation in any detail because it is outside the method of production of food regulated by EU Reg. No. 2017/625.   This is a lacuna in the narrative that requires addressing.

  1. Ferrere MR, King M, and Larsen LM. Animal Welfare in New Zealand: Oversight, Compliance and Enforcement. Printed by Uniprint, University of Otago: New Zealand Law Foundation, 2019.
  2. Morris MC. The use of animals in New Zealand: regulation and practice. . Society & Animals 2011;19 (4): 368-382.
  3. Shroff V. Superior court reins in OSPCA’s police-type powers. The Lawyer's Daily 2019.

Author Response

Reviewer: From page 1-9 the authors are describing “animal welfare” as conceived by EU Reg. No. 2017/625 which deals with animals that will become food. So for the majority of the paper where the “animal welfare" appears it can be read as “food animal welfare”. At page 9 . Line 350-367 the subject of “pets” and dogs and cats are included in the narrative. After line 369 we are back to food animals.
Our response: The paper’s focus on food animal welfare is now stated and explained in the first paragraph of the introduction on page 2 (line 51-55)

Reviewer: I would request a paragraph about line 70 that described the quazi-criminal legislation protecting companion animals in the two countries, who provides that policing and how this paper will not consider this protection of animals legislation in any detail because it is outside the method of production of food regulated by EU Reg. No. 2017/625.   This is a lacuna in the narrative that requires addressing.
Our response: The issue of companion animals is dealt with in a new (second) paragraph on page 2 (line 66-77). Also, the paper’s focus on food animal welfare in stated in the beginning of the first paragraph of the introduction on page 2.